# Elucidating Uncertainty in Heat Vulnerability Mapping: Perspectives on Impact Variables and Modeling Approaches

**DOI:** 10.3390/ijerph21070815

**Published:** 2024-06-21

**Authors:** Sockho Jeong, Yeonyeop Lim, Yeji Kang, Chaeyeon Yi

**Affiliations:** 1Chungnam Economy Promotion Agency, Support Center for Community Revitalization, Yesan-gun 32416, Republic of Korea; sockhoj@cepa.or.kr; 2Department of Geography, Kongju National University, Kongju-si 32588, Republic of Korea; siawase@kongju.ac.kr; 3Korea Adaptation Center for Climate Change, Korea Environment Institute, Sejong-si 30116, Republic of Korea; yjkang@kei.re.kr; 4Research Center for Atmospheric Environment, Global Campus, Hankuk University of Foreign Studies, Yongin-si 17035, Republic of Korea

**Keywords:** heat vulnerability map, uncertainty, heatwave impacts, mortality, morbidity

## Abstract

Heat vulnerability maps are vital for identifying at-risk areas and guiding interventions, yet their relationship with health outcomes is underexplored. This study investigates the uncertainty in heat vulnerability maps generated using health outcomes and various statistical models. We constructed vulnerability maps for 167 municipalities in Korea, focusing on the mild and severe health impacts of heat waves on morbidity and mortality. The outcomes included incidence rates of heat-related outpatient visits (morbidity) and attributable mortality rates (mortality) among individuals aged 65 years and older. To construct these maps, we utilized 11 socioeconomic variables related to population, climate, and economic factors. Both linear and nonlinear statistical models were employed to assign these socioeconomic variables to heat vulnerability. We observed variations in the crucial socioeconomic variables affecting morbidity and mortality in the vulnerability maps. Notably, nonlinear models depicted the spatial patterns of health outcomes more accurately than linear models, considering the relationship between health outcomes and socioeconomic variables. Our findings emphasize the differences in the spatial distribution of heat vulnerability based on health outcomes and the choice of statistical models. These insights underscore the importance of selecting appropriate models to enhance the reliability of heat vulnerability maps and their relevance for policy-making.

## 1. Introduction

Climate change has induced a surge in extreme weather events worldwide, including heatwaves, cold waves, typhoons, floods, and droughts [1]. Heatwaves have become increasingly frequent in Europe, Australia, and South Korea, primarily due to the compounding effects of global warming and urbanization [2,3]. Numerous regions of South Korea have experienced unprecedented temperature spikes since 2018, marking a significant departure from historical meteorological records [4].

The adverse health effects of heat waves have been extensively documented in various countries. For instance, in 2003, approximately 72,210 deaths were attributed to heatwaves across 15 European nations [5]. Similarly, the UK reported 892 excess deaths during a heatwave in 2019, while [6], Portugal and Spain experienced 1,063 and 679 excess deaths, respectively, during heatwaves in June and July 2022 [7]. South Korea also faced a grim toll, with an estimated 800–900 excess deaths attributed to heatwaves in 2018 [8,9].

Future projections indicate that climate change and urbanization will exacerbate the intensity, frequency, and duration of heatwaves [1]. Given the profound implications of heatwaves on human health, understanding the regions susceptible to heatwaves is of paramount importance for public health and safety. Effective response strategies and protective measures hinge on the accurate identification of vulnerable areas.

Heat vulnerability maps serve as invaluable tools for identifying regions vulnerable to heat waves. By visually highlighting areas with heightened vulnerability, these maps can assist policymakers in devising targeted intervention measures. Additionally, they raise awareness of health-related issues and aid public health organizations in formulating appropriate response strategies. Moreover, heat vulnerability maps can be used to anticipate spatial pattern changes in vulnerability, rendering them indispensable for urban planning and sustainable development.

Previous studies on heat vulnerability maps have predominantly focused on analyzing regional vulnerability using socioeconomic variables. Various methodologies, such as principal component (PC) analysis and equal weighting assumptions, have been employed for variable selection and weighting [10,11,12,13,14,15]. Research on the uncertainty of heat vulnerability maps is of great interest to researchers. Some studies have shown that caution is required when relying solely on heat vulnerability maps constructed using a single method compared to those constructed using different methods [16,17].

However, there remains a dearth of studies exploring the uncertainty surrounding the explanatory power of health outcomes in heat vulnerability maps. Health outcomes characterize heat-related outpatient visits as a mild impact, and emergency room admissions and mortality as a severe impact [18]. Several studies have compared the explanatory power of health outcomes and heat vulnerability maps but only used mortality as an indicator [17,19]. Most other studies have not attempted to compare health outcomes and heat vulnerability maps. Therefore, little is known about the relationship between heat vulnerability maps and health outcomes. For instance, it is difficult to determine which heat impact indicator’s spatial distribution is better explained by heat vulnerability maps and whether the same variables can be used to construct vulnerability maps for mild and severe health outcomes. Understanding the relationship between health outcomes and heat vulnerability maps is essential for identifying vulnerability factors and developing policies. Differences in socioeconomic vulnerability factors, depending on the impact indicator, could provide critical information for policy development.

Previous studies have assumed a linear relationship between independent variables and health outcomes and constructed heat vulnerability maps through a combination of independent variables. However, the possibility of a nonlinear expression of the influence of specific variables due to interactions between independent variables and the existence of thresholds in the influence of independent variables on dependent variables has been overlooked. 

Additionally, the impact of heatwaves varies depending on socioeconomic conditions such as age, occupation, and income, with older adults being well known as a group vulnerable to heatwaves [20,21,22]. Limited mobility and chronic illnesses (e.g., cardiovascular diseases) of the elderly may affect their ability to adequately protect themselves and respond promptly to extreme weather [23]. Furthermore, their acquisition and utilization of information related to heatwaves are limited [24,25].

The increasing number of elderly people vulnerable to heatwaves could exacerbate the societal and economic risks of heatwaves, leading to significant social problems. South Korea is one of the countries experiencing rapid aging globally, with the proportion of the elderly population increasing from 7.2% in 2000 to 16.6% in 2021 [26]. Sim et al. calculated the exposure risk of the population to heatwaves based on future scenarios of heatwave days and national population estimates, forecasting that rapid aging would be a significant factor in increasing the exposure risk to heatwaves [27]. An increase in the elderly population in the future is expected to escalate the scale of heatwave damage alongside an increase in extreme weather phenomena [28]. Therefore, despite several studies, continued interest and research on the elderly are needed.

The HVI (Heat Vulnerability Index) mapping can be used as a proxy measure for potential health impacts when comprehensive health data is not available. Furthermore, it provides various insights that are difficult to identify from simple health impact indicators, offering advantages in policy development and future risk prediction. HVI mapping takes a multifaceted approach by integrating demographic, socioeconomic, and climate variables instead of relying on a single indicator or variable. This allows for a more comprehensive assessment of heat vulnerability in specific regions, enabling the analysis of key factors contributing to regional vulnerability. This approach offers a more holistic understanding compared to analyzing individual health outcomes alone. It provides policymakers with the information necessary for developing tailored policies and effectively allocating resources to address socioeconomic weaknesses in specific regions. Additionally, while direct health impact indicators show the current state or past impacts, HVI can help predict potential future risks. This plays a crucial role in minimizing health impacts through preventive measures. Therefore, the approach to improving HVI mapping proposed in this study will play a vital role in predicting future risks, developing tailored policies, and enhancing preventive measures.

The objective of this study was to enhance the understanding of the uncertainties that need to be considered when constructing heat vulnerability maps. First, we aimed to examine the relationship between heat vulnerability outcomes and health outcomes, proposing a method for combining heat vulnerability outcomes based on linear and nonlinear relationships. Furthermore, this study aimed to construct heat vulnerability maps based on the nonlinear relationship between heat vulnerability and health outcomes. Finally, we aimed to investigate differences in socioeconomic vulnerability factors associated with various health outcomes of heatwaves, thereby providing critical insights for policy development.

## 2. Materials and Methods

### 2.1. Data

The spatial scope of this study encompassed 167 municipalities in South Korea (Figure 1). The analysis covered the period from 2007 to 2020, spanning 14 years. Two negative health outcomes of for heatwaves were considered: morbidity, representing a mild impact, and mortality, representing a severe impact. Morbidity was measured using the incidence rate of heat-related illnesses during the summer months (June–August), whereas mortality was assessed using the proportion of excess deaths attributable to heatwaves.

Data on outpatient visits for heat-related illnesses were obtained from the National Health Insurance Service of South Korea, utilizing healthcare big data. Heat-related illnesses (HRI) were defined using the T67 code from the 10th Revision of the International Classification of Diseases (ICD-10). Mortality data were acquired from Statistics Korea’s cause-of-death statistics, including all deaths from external and other causes, and extracted using ICD-10 codes A-R. In this study, older adults were defined as individuals aged 65 and above. The number of outpatient visits for heat-related illnesses among older adults during the summer months was 70,685, and the total number of older adult deaths over the entire period was 4,006,289.

A total of 11 socioeconomic variables were used to create the heat vulnerability map. A list of variables and a map for each variable can be seen in Table 1 and Figure 1, respectively. Population variables include factors related to the elderly population, occupation, and household structure. The variables included the proportion of the population aged 65 or older (pop_65), the proportion of the population in rural areas (agri_65), and the proportion of the population living alone (single_65).

Economic variables include factors that reflect the income level of the elderly population and social infrastructure. The income level of the elderly population (income_65) was calculated as the ratio of low-income households among the population aged 65 or older. For low-income households, data from the lower insurance premium class (bottom 20%) were used. Financial independence refers to the ratio of self-revenue, which is the basis of a local government’s independent financial capacity, to its budget size. It is an indicator showing the degree of self-reliance based on each local government’s internal revenue, and the higher the ratio, the better the self-reliant basis of the local government.

For climate variables, temperature and humidity data from each region from 2007 to 2020 (14 years) were used to explain exposure to heat waves. The 90th percentile temperature (t90) of each region represents the level of high temperature throughout the year. The number of days exceeding the 95th percentile (tmax95) indicates the number of days when the daily maximum temperature in summer was above the 95th percentile, reflecting the intensity and frequency of heat waves. The daily average temperature (tag) in summer for each region reflects the average summer temperature.

Summers in Korea tend to have high humidity, which is closely related to the occurrence of heat-related illnesses. Therefore, we included humidity data in our analysis. The number of days (rhav80) with relative humidity above the 80th percentile in summer for each region indicates the intensity and frequency of high humidity. Additionally, the average summer relative humidity (rhav_avg) of each region was included as a variable.

We adopted the daily maximum temperature for our study. Although some existing studies have used average temperature [29], those analyses included the influence of winter. In contrast, this study focuses solely on the summer period. Several studies in Korea have demonstrated the health effects of the daily maximum temperature, and the heatwave warning system has long been based on this metric [18,22]. Therefore, our analysis was performed using the daily maximum temperature.

Furthermore, the ratio of forest to total area was used as a proxy variable for assessing the impact of green space on alleviating the thermal environment.

Temperature and humidity data, consisting of climate variables, were obtained from the Automated Synoptic Observation System of the Korea Meteorological Administration. Area average data for each region were assigned using the Thiessen polygon method. Commonly used interpolation methods for meteorological applications include nearest-station assignment, inverse-distance weighting, inverse-distance-square weighting, Thiessen polygon method, orthogonal-polynomial approximation, Lagrange method, interpolation by splines, kriging, and interpolation by empirical orthogonal functions. Each method has its merits and is applicable based on the temporal scale, spatial scale, stationarity, and variability of the field under consideration [30].

The Thiessen polygon approach determines the average weather by considering only the representative area of a station, which is determined by the bisector between each pair of stations. Although this method has limitations, such as not accounting for topographic effects, it is useful for producing long-term regional average weather data [31].

### 2.2. Ethical Approval

The study protocol underwent thorough review and approval by the Institutional Review Board (IRB) of Kongju National University (Confirmation No. 2022-46). Given the observational nature of this study and the use of anonymized statistical data, the IRB committee waived the requirement for informed consent. All methodologies adhered to the guidelines stipulated by the Korean government for health and medical data usage.

### 2.3. Methodology for Impact Indicator Calculation

The attributable mortality rate of excess deaths was calculated based on the statistical relationship between the daily maximum temperature and mortality. Initially, the relationship between the daily maximum temperature and mortality was analyzed regionally. The Distributed Lag Non-Linear model (DLNM) was adopted for this purpose. The DLNM, known for its ability to account for the lag effects of independent variables, has been widely utilized in recent epidemiological research [22]. The model was formulated as shown in Equation (1), and analysis was conducted using R version 4.3.0 with the dlnm package.
(1)$$In\left(y_{ij}\right)=\alpha +\beta Tmax_{ij,jl}+NS\left(sn_{ij}\right)+NS\left(rhavg_{ij}\right)+NS\left(doy_{ij}\right)+\gamma weekday_{ij}+\varepsilon$$
where *Y_ij_* represents the number of deaths recorded in region *j* on day *i*. *Tmax* denotes the daily maximum temperature, and *l* represents the lag effect. Considering the tendency for the impact of heatwaves to be strongest on the same day and rapidly decrease within a short period [32,33,34], we set the lag period to three days. Here, Rhavg refers to the average daily humidity, *doy* represents the day of the year in terms of Julian days, and *sn* denotes the serial number. The degree of freedom for this variable was set to 6 and multiplied by 14 years (the duration of the study data). Weekdays indicate the days of the week. Natural cubic splines (*NS*) were used to examine the non-linear relationship between the dependent and independent variables. Based on the relative risk of death calculated from temperature using the aforementioned method, excess deaths for each region were calculated using Equation (2) [8].
(2)$$Y_{ij}=D_{ij}\times \frac{e^{\beta \times \Delta T_{ij}}-1}{e^{\beta \times \Delta T_{ij}}}\times H_{ij}$$
where *i* and *j* represent the variables for the date and region, respectively, *Y_ij_* denotes the number of excess deaths that occurred in region *j* on day *i*. *D* represents the daily deaths, and *β* indicates the slope of the relative risk above the threshold temperature. The threshold temperature for region *j* was determined by observing the non-linear relationship between the temperature and deaths mentioned above. Δ*T* represents the difference between the threshold and the temperature for each day. If the temperature in region *j* on day *i* exceeded the threshold temperature, *H* was 1; otherwise, it was 0.

The contribution rate of excess deaths for each region was calculated by averaging the percentage of excess deaths compared to the total number of deaths per year. The heat-related illness incidence rate was calculated by averaging the number of heat-related illness cases per ten thousand elderly population per year.

Throughout all stages of the analysis, including the calculation of excess death contribution rates, heat-related illness incidence rates, and socioeconomic variables, data for seven metro cities (Seoul, Busan, Daegu, Incheon, Gwangju, Daejeon, and Ulsan) were exclusively used for each city. However, for other cities and rural areas, data were integrated from adjacent regions because the sample size for these areas was sometimes inadequate due to low variability and limited data availability to establish statistical models. Therefore, to ensure an adequate sample size for the analysis, data from adjacent regions were integrated. Consequently, the interpretation of the results considered these areas from a regional perspective, rather than focusing on individual cities outside metro areas.

### 2.4. Methodology for Analysis of Relationships between Variables

The analysis of the relationships between health outcomes and socioeconomic variables proceeded as follows: Firstly, the socioeconomic variables were transformed into components without multicollinearity issues using a PC analysis. Subsequently, these components were utilized as independent variables, while the relative risks of morbidity and mortality were employed as dependent variables to analyze their relationships. Both linear and nonlinear models, represented by Equations (3) and (4) respectively, were used in this analysis.
Y = β_0_ + β_1_X_1_ + β_2_X_2_ + … + β_n_X_n_ + ϵ(3)
Y = β_0_ + f_1_(X_1_) + f_1_(X_2_) + … + f_n_(X_n_) + ϵ(4)
where Y represents the dependent variable, which can be either the incidence rate of heat-related illnesses or the attributable mortality rate. β_0_ denotes the intercept of the model, while X_1_–X_n_ represents the independent variables. β_1_–β_n_ are the coefficients for each independent variable. f_1_–f_n_ represents the smoothing functions for each independent variable to account for nonlinearity. For linear and nonlinear models, we used R’s linear model (LM) and the generalized additive model (GAM), respectively.

### 2.5. Methodology for Generating Heat Vulnerability Maps

Heat vulnerability maps were constructed using two approaches. The first was based on the results of a linear model. Utilizing the linear relationship analysis mentioned earlier, vulnerability maps were created using PCs that exhibited statistically significant relationships (*p* < 0.05) with health outcomes (incidence rate of heat-related illnesses and attributable mortality rate) at a 95% confidence level. The PC values were assigned to nine risk levels, with the direction of risk allocation determined based on the relationship (negative or positive) between the PCs and the health outcomes. The final heat vulnerability map was constructed by averaging the maps assigned to the PCs. In this study, we refer to this method as the Heat Vulnerability Index based on the Linear Model (HVI-LM).

The second approach was based on the results of a nonlinear model (NLM). This map was created based on previously analyzed nonlinear relationships. Firstly, we identified the thresholds in the nonlinear relationship between the health outcomes and PCs. The changes observed above these thresholds were used to assign the values of the PCs to the nine risk levels. We refer to this method as the Heat Vulnerability Index based on the Nonlinear Model (HVI-NLM).

## 3. Results

### 3.1. Health Outcoms of Heatwave

The incidence rate of heat-related illnesses and the attributable mortality rate from heat waves exhibited spatial variation. The incidence of heat-related illnesses was generally higher in the southwestern region of Korea and lower in the northeastern region (Figure 2a). In contrast, the areas with higher attributable mortality rates were dispersed across various regions (Figure 2b). Due to the complexity of the spatial distribution of mortality compared to that of diseases, it may be difficult to statistically explain the spatial distribution of mortality. This observation reinforces the findings of the analysis of the relationship between health outcomes and PCs.

### 3.2. Principal Components

PC analysis of the socioeconomic variables revealed four PCS, named PC1, PC2, PC3, and PC4. Table 2 summarizes the results of the PC analysis conducted using the 11 variables. Four PCs showed initial eigenvalues exceeding 1, and the cumulative explained variance for the first four components was 86.08%. 

Table 3 lists the weights of the variables for the four PCs. Based on this table, we identified the key variables of each PC and defined the nature of each component. PC1 represented the demographic characteristics of the elderly population in the area and socioeconomic conditions. This component consisted of variables such as the proportion of the elderly population (pop_65), the proportion of the elderly engaged in agriculture (agri_65), the proportion of the elderly living alone (single_65), and the financial autonomy of the local government (financial). PC2 represented the climatic component of the area, comprising temperature (t90 and tmax95) and humidity variables (rhav80 and rhav_avg). PC3 consisted of the cover ratio of the forest (per_forest), and average temperature (tavg), defining a combination of temperature and a variable that can moderate its effects. Lastly, PC4 was composed of the low-income elderly population ratio (lowincome_65) and temperature variable, defining a component of the economic conditions of the elderly and the possibility of exposure to heatwaves.

### 3.3. Relationship between PCs and Impacts

We derived three key findings regarding the relationship between the PCs and health outcomes. First, the NLM showed higher statistical explanatory power for the spatial distribution of health outcomes than the linear model. Table 4 summarizes the *p*-values of the PCs in the analysis of their relationships with the health outcomes. The R^2^ values of the LM were 0.381 and 0.154 for morbidity and mortality, respectively. In contrast, the R^2^ values for the NLM were 0.518 and 0.486 for morbidity and mortality, respectively. The NLM performed better than the LM in terms of both morbidity and mortality. Secondly, the second PC explained morbidity better than mortality. For LM, the R^2^ value for morbidity was 0.381, which was higher than that for the mortality rate of 0.154. Similarly, in the NLM, the R^2^ for morbidity was 0.518, compared to 0.486 for mortality. In the NLM, the explained deviations were 55.7% for morbidity and 52.5% for mortality. Thirdly, statistically significant PCs varied between morbidity and mortality. PC1 was significantly associated with morbidity, whereas PC4 was significantly associated with mortality. PC3 was a statistically significant variable in all analyses.

In the LM model, PC2 had a negative relationship, while PC3 had a positive relationships with the health outcomes. The linear regression slopes of PC2 for morbidity and mortality were −1.1151 and −0.1113, respectively. On the other hand, the linear regression slopes for PC3 in terms of morbidity and mortality were 3.1630 and 0.1010, respectively. In the case of morbidity, the slope of PC1 was 2.0553, while in the case of mortality, the slope of PC4 was −0.1414.

Similar relationships between health outcomes and PCs were observed in the NLM (Figure 3). PC1 exhibited a nonlinear relationship with morbidity, showing an increasing trend in morbidity when PC1 was not significant in mortality. In morbidity, PC2 and PC3 had statistically significant negative and positive correlations, respectively. PC3 did not show a statistically significant relationship with morbidity. Mortality showed a nonlinear relationship with PC3 and PC4. PC1 and PC2 did not show a statistically significant relationship with mortality. PC3 exhibited a positive relationship with mortality. PC4 served as a threshold point at approximately −1, with no statistically significant change in mortality above −1. However, mortality increased distinctly below −1.

### 3.4. Comparison between HVI-LM and HVI-NLM

Based on the analysis results, the values of the PCs were assigned to a heat vulnerability index (HVI) in nine intervals ranging from −4 to 4 (Table 5). Each interval was considered using probabilities of 1%, 5%, 10%, 25%, 50%, 75%, 90%, 95%, and 99% after standardizing the PCs using means and standard deviations. When assigning the PCs based on the results of HVI-LM, if the linear regression coefficient was positive, the values of each PC were sequentially assigned from −4 to 4. Conversely, if the linear regression coefficient was negative, the values of each PC ranged from 4 to −4. For the HVI-NLM, the HVI was assigned only to the intervals of PCs where a relationship between the dependent and independent variables was observed.

For instance, in a nonlinear relationship, when PC1 exceeded approximately 1, the incidence rate of heat-related illnesses showed a dramatic increase. When PC1 was near 0, no statistically significant change was recorded in the incidence rate of heat-related illnesses. Therefore, the HVI gradually increased from the negative extreme value of PC1 to the positive values. However, the HVI was not assigned around 0 for PC1. Unlike linear relationships, assigning an HVI of 0 for intervals where statistically significant relationships were not observed in nonlinear relationships could result in excessive smoothing of the final results because the HVI was averaged for all PCs.

After assigning the HVI based on the range of each PC, a heat vulnerability map was created by averaging the HVIs of the PCs (Figure 4). The results showed significant differences in the spatial distribution of heat vulnerability predictions depending on the impact of heat waves (morbidity and mortality) and vulnerability assignment methods (linear and nonlinear). In the heat vulnerability maps for morbidity and mortality, areas with high vulnerability to morbidity were concentrated in the southwest region, whereas vulnerability to mortality was generally observed in coastal areas such as the western and eastern coastal regions.

Furthermore, even when the same PC was used, distinct differences were observed between (a) HVI-LM-morbidity and (c) HVI-NLM-morbidity, as well as between (b) HVI-LM-mortality and (d) HVI-NLM-mortality. In (a) HVI-LM-morbidity, vulnerable areas were concentrated in the southwest region, whereas in (c) HVI-NLM-morbidity, vulnerable areas included broader inland areas in the west and southeast compared with (a). In (b) HVI-LM-mortality, vulnerability in the northeast region was classified as low, whereas in (d) HVI-NLM-mortality, vulnerability was classified as high. Differences were also noted in the vulnerability between the two maps in the southeastern region.

## 4. Discussion 

This study demonstrated significant differences in heat vulnerability maps based on health outcomes and statistical models. The results revealed notable variations in socioeconomic variables that were statistically significant for morbidity and mortality risks. For morbidity, PC1 (size of the elderly population and socioeconomic conditions) and PC3 (temperature and the effect of green space) showed statistically significant explanatory power. In contrast, PC2 (climate conditions) and PC4 (income of the elderly and temperature) demonstrated statistically significant explanatory power for mortality. These differences contributed to variations in the heat vulnerability maps for morbidity and mortality.

Furthermore, this study identified nonlinear relationships between socioeconomic variables and health outcomes. Several existing studies that have constructed heat vulnerability maps [13,14,15,16,17] have assumed a linear relationship between socioeconomic variables and vulnerability, assigning vulnerability based on changes in socioeconomic variables. However, our study indicates that NLMs better explain the relationship between socioeconomic variables and health outcomes than LMs. Heat vulnerability maps based on nonlinear relationships matched the spatial patterns of impacts better than those using linear models.

The implications of these findings underscore the importance of considering health outcomes and statistical modeling factors when constructing heat vulnerability maps. Given the utilization of heat vulnerability maps as a tool for formulating heat response policies, differences in health outcomes have significant implications. A heat vulnerability map that does not specifically identify heat-related health risks may have limitations when utilized in policymaking. Regions with high morbidity risks may require education and preventive measures to raise awareness about heat-related illnesses, whereas regions with high mortality risks may require strengthening of healthcare facilities, emergency services, and support for vulnerable populations, such as low-income individuals and the disabled. Therefore, ensuring clarity in health outcomes when developing heat vulnerability maps can enhance policy utility.

Previous studies have not addressed the various types of damages that can be identified using heat vulnerability maps. According to prior research, because the temperature thresholds for morbidity and mortality differ, different response policies are required in terms of timing and risk communication. Since morbidity tends to precede mortality, studies have indicated challenges in providing adequate warnings for a large number of morbidity cases when heat warning systems are primarily designed to address mortality [18,35]. Similarly, the distinct distribution of heat vulnerability for morbidity and mortality, along with differences in the PCs associated with mortality and morbidity, implies the need for tailored response policies for each region. Therefore, constructing separate heat vulnerability maps for morbidity and mortality could assist in formulating customized policies for each region.

However, this study has certain limitations that highlight the need for further investigation. First, additional research is required to enhance the explanatory power of the spatial distribution of heat-related damage. The explanatory power for morbidity and mortality in this study was 55.7% and 52.5%, respectively, indicating that there is room for improvement. This could involve exploring dynamic data that reflect societal activities. In an existing case targeting a single city, the coefficient of determination between the heat vulnerability index and heat-related disease deaths was 0.58 using an improved method and 0.32 using the existing method [16]. This suggests that it is challenging to accurately reproduce actual impacts with the heat vulnerability index. The selection of different variables can lead to significantly different results on heat vulnerability maps. Therefore, efforts to develop variables and find appropriate combinations are crucial [36].

For example, there is a need to develop variables for social ties [37], effective heat wave warning systems [24,38,39] and the quality and accessibility of medical services [40]. Additionally, variables representing the operational and activity levels of social and cultural facilities should be considered. Cultural facilities not only serve as shelters from heat but also provide gathering spaces and activities. In Seoul, for example, cultural facilities extend their operating hours during the summer and offer various programs. The operations of cultural facilities encompass structural (installation, development, and maintenance), nonstructural (investment, support, and monitoring), and social (collaboration, education, promotion, and guidelines) measures. Quantifying regional differences in such dynamic variables can be incorporated into the construction of heat vulnerability maps.

Additionally, in terms of modeling, it is necessary to apply the NLMs identified in this study while determining the extent to which variables should be minimized or maximized to enhance the model’s explanatory power. Finding the appropriate balance is crucial. Although the integration of factor analysis and DLNM used in this study was essential for minimizing variables and identifying statistically significant causal relationships, the DLNM model, which reduces the variance of model estimates in a nonlinear fashion, poses challenges in interpreting the results. Future research is required to enhance the overall reliability of DLNM and select significant factors that can differentiate between urban and non-urban areas.

Secondly, when using NLMs, it is necessary to establish a more systematic process for assigning heat vulnerabilities. The finding that NLMs yield better results than LMs is significant. However, in NLMs, the assignment of vulnerability values based on researcher observations introduces the potential for subjective judgment, which can hinder the construction of heat vulnerability maps based on NLMs. To address this issue, it is necessary to accumulate various research cases based on NLMs and establish a systematic assignment process.

Despite these challenges, this study identified the impact variables associated with heat-related morbidity and mortality and revealed the spatial patterns of heat vulnerability for morbidity and mortality at the municipal level. Addressing the limitations and conducting further research will contribute to the refinement and effectiveness of heat vulnerability mapping.

## 5. Conclusions

This study highlights the differences in heat vulnerability maps based on health outcomes and statistical models, underscoring the significant impact of socioeconomic variables on morbidity and mortality risks. Our findings indicate that the variables significantly associated with morbidity and mortality differed. Additionally, non-linear models offered a more accurate representation of the relationship between socioeconomic variables and health outcomes compared to linear models. These results emphasize the importance of selecting appropriate statistical models to enhance the accuracy of heat vulnerability maps and the need to configure variables differently depending on the specific health outcomes.

Several limitations were identified in the heat vulnerability mapping process. Firstly, there is the issue of low explanatory power. Addressing this challenge requires efforts to develop various variables beyond those proposed in this study. Variables such as social cohesion, effective heat warning systems, and the quality and accessibility of healthcare may be crucial. Furthermore, research is needed to minimize subjective judgment when using nonlinear models. To enhance model reliability, establishing a systematic process for assigning the results of nonlinear models to heat vulnerability is essential.

In conclusion, our study underscores the importance of considering both health outcomes and statistical modeling factors in constructing heat vulnerability maps. These maps serve as crucial tools to inform heat response policies and facilitate targeted interventions based on specific regional vulnerabilities. Insights from this study can assist national and local governments in prioritizing areas for heat response and management. Additionally, our findings offer valuable guidance for researchers developing heat vulnerability maps and advocate for the integration of advanced statistical modeling techniques with health outcomes.

## Figures and Tables

**Figure 1 ijerph-21-00815-f001:**
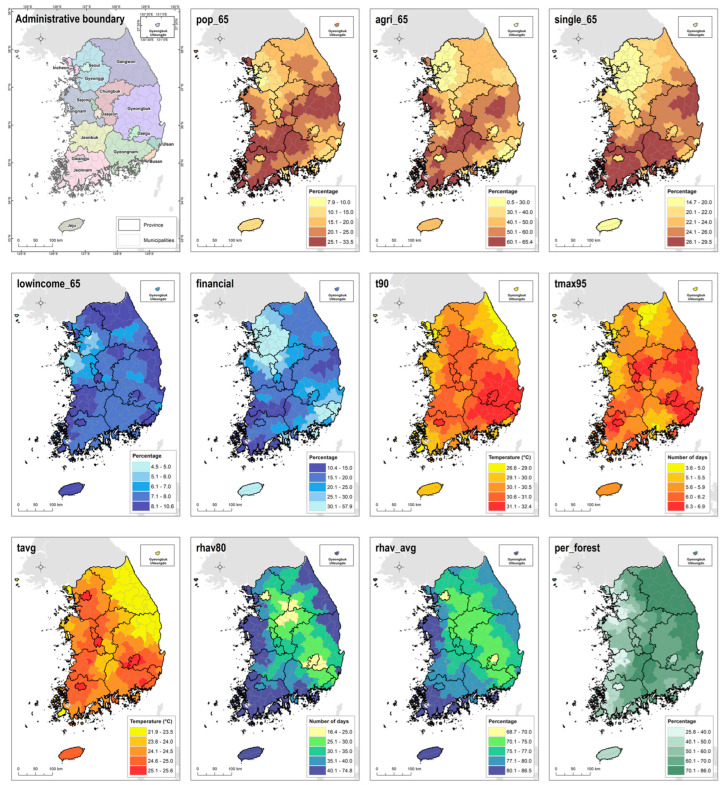
Socioeconomic maps used in the study.

**Figure 2 ijerph-21-00815-f002:**
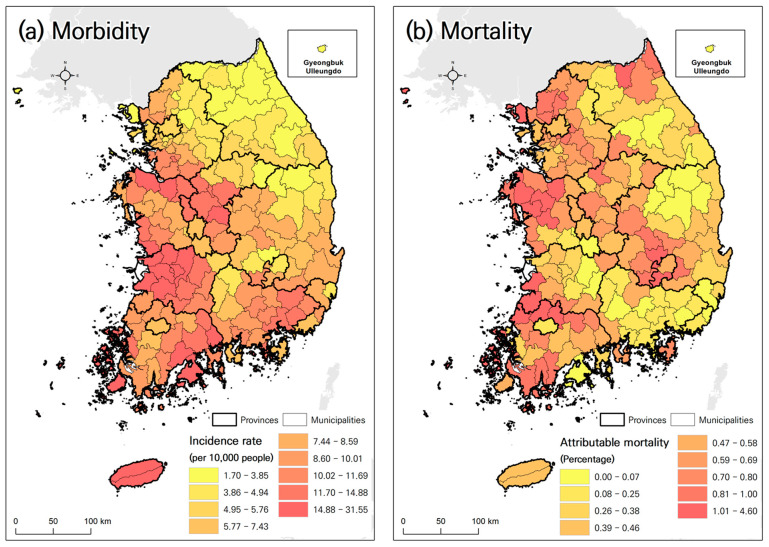
Spatial pattern of health outcomes in Korea (2007–2020).

**Figure 3 ijerph-21-00815-f003:**
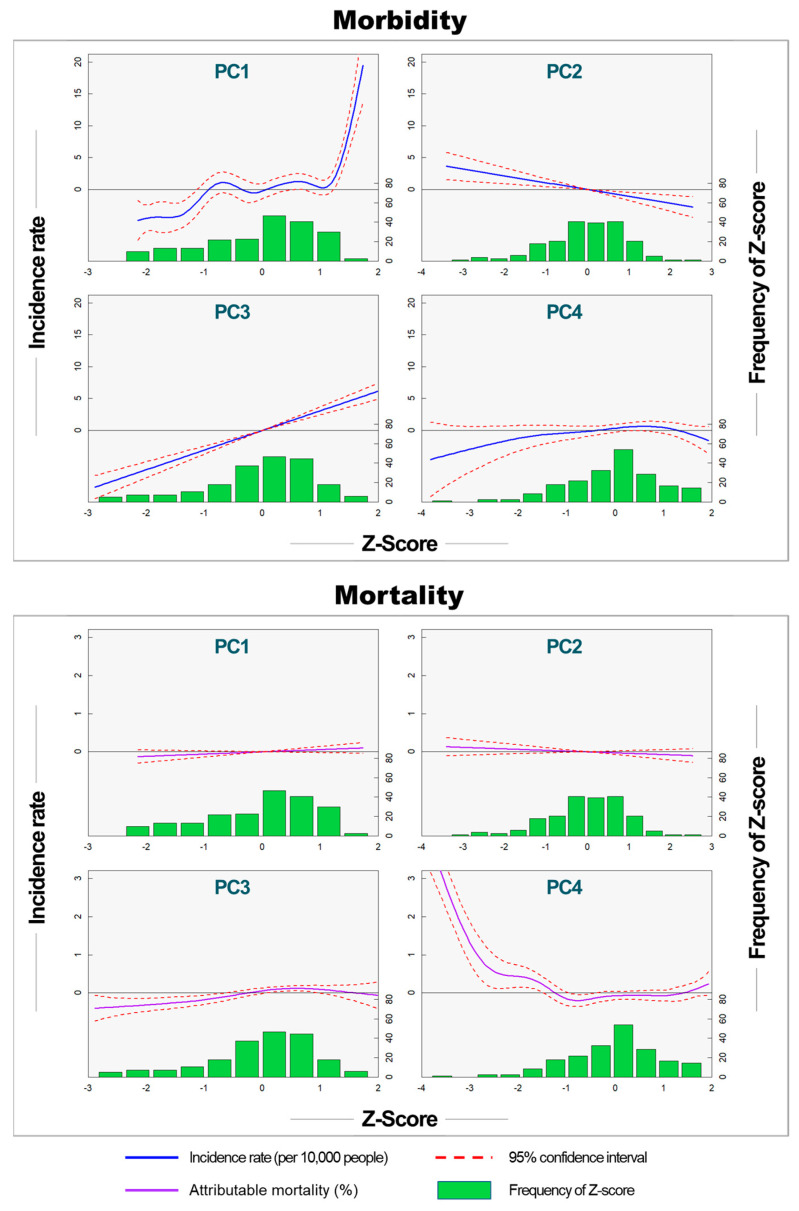
Nonlinear relationship between Z-scores of PCs and morbidity and mortality (Solid blue lines represent the morbidity and mortality contribution rates. The dashed blue lines represent the 95% confidence level).

**Figure 4 ijerph-21-00815-f004:**
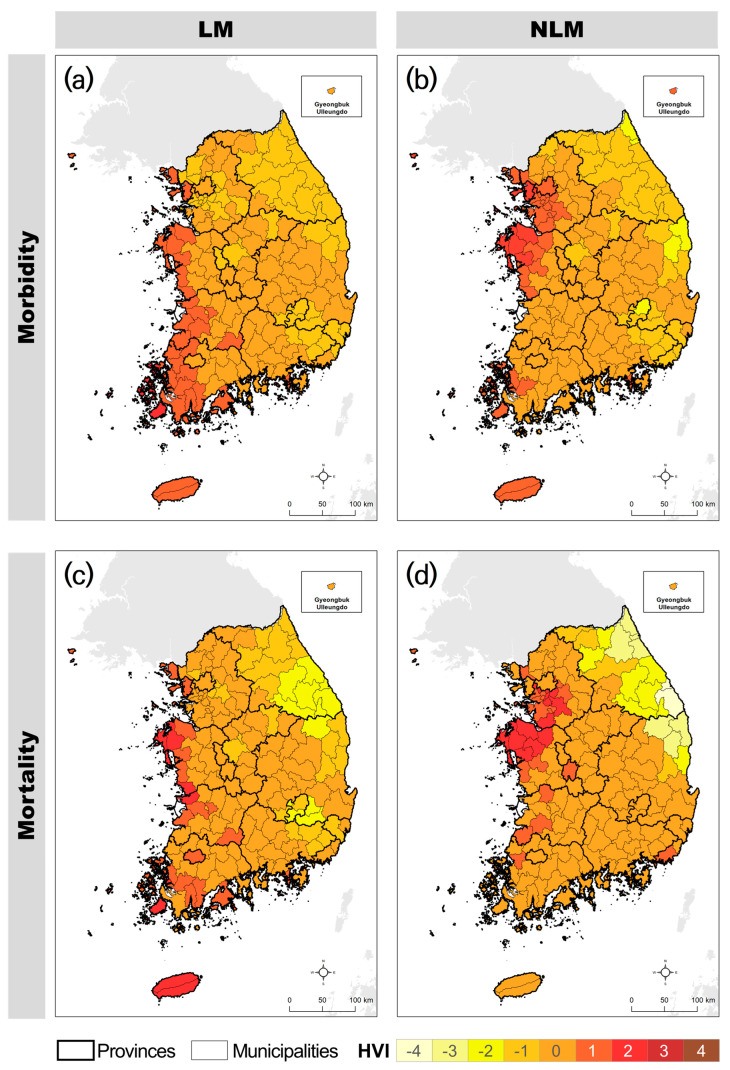
Comparison of heat vulnerability indices.

**Table 1 ijerph-21-00815-t001:** Socioeconomic variables considered for constructing heat vulnerability maps in the study.

Type	Variable	Description	Mean	Range	Period
Population	pop_65	Population aged 65 and higher (%)	19.6	7.9–33.5	2007–2020
	agri_65	Percentage of agricultural workers among the population aged 65 and higher (%)	41.9	0.5–65.4	2007–2020
	single_65	Percentage of single-person households among the population aged 65 and higher (%)	22.8	14.7–29.5	2010–2020
Economic	lowincome_65	Percentage of low-income households among the population aged 65 and higher (%)	7.7	4.5–10.6	2010–2020
	financial	Municipal fiscal self-reliance (%)	25.1	10.4–57.9	2007–2020
Climate& Environment	t90	90^th^ percentile temperature (°C) for each region (2007–2020)	30.3	26.6–32.4	2007–2020
	tmax95	Number of days with temperatures above the 95^th^ percentile (2007–2020) compared to those with 90th percentile temperature (°C) in summer (JJA) from 1999 to 2020	5.8	3.6–6.9	2007–2020
	tavg	Average temperature (°C) in summer	24.2	21.9–25.6	2007–2020
	rhav80	Number of days with relative humidity exceeding 80% in summer (JJA)	38.7	16.4–74.8	2007–2020
	rhav_avg	Average relative humidity (%) in summer (JJA)	77.0	68.7–86.0	2007–2020
	per_forest	Forest occupies the total area (%)	60.0	25.8–86.0	2007–2020

**Table 2 ijerph-21-00815-t002:** Summary of principal component analysis results.

Component	Initial Eigenvalues		
	Total	% of Variance	Cumulative
1	4.32	39.29	39.29
2	2.85	25.94	65.23
3	1.63	14.81	80.04
4	1.96	8.68	88.72
5	0.63	5.69	94.41
6	0.25	2.24	96.65
7	0.15	1.37	98.02
8	0.11	0.99	99.01
9	0.05	0.50	99.51
10	0.04	0.38	99.89
11	0.01	0.11	100.00

**Table 3 ijerph-21-00815-t003:** Matrix of the first four principal components.

Variable	PC1	PC2	PC3	PC4
pop_65	0.906	0.238	0.160	−0.191
agri_65	0.844	0.236	0.227	−0.314
single_65	0.863	0.303	0.236	0.015
lowincome_65	0.408	0.200	−0.209	0.804
financial	−0.927	−0.185	−0.025	−0.077
t90	−0.272	0.832	0.395	−0.064
tmax95	0.036	0.665	0.148	0.321
tavg	−0.516	0.331	0.739	0.059
rhav80	0.452	−0.799	0.330	0.118
rhav_avg	0.500	−0.743	0.367	0.109
per_forest	0.471	0.374	−0.697	-0.171

The colored areas are where high impact occurred.

**Table 4 ijerph-21-00815-t004:** *p*-values of principal components in the regression models.

Model	Outcome	PC1	PC2	PC3	PC4	Adjusted R^2^
LM	Morbidity	**<0.001**	**<0.001**	**<0.001**	0.155	0.381
	Mortality	0.078	**0.00** **6**	**0.** **015**	**0.00** **1**	0.154
NLM	Morbidity	**<0.001**	**0.001**	**<0.001**	0.184	0.518
	Mortality	0.150	0.264	**<0.001**	**<0.001**	0.486

Statistically significant parts are marked in bold.

**Table 5 ijerph-21-00815-t005:** Assigning values of the heat vulnerability index.

Method	Impact	Range of Components	Assigned HVI			
			PC1	PC2	PC3	PC4
HVI-LM	Morbidity	<−2.326	−4	4	−4	
		−2.326–−1.645	−3	3	−3	
		−1.645–−1.282	−2	2	−2	
		−1.282–−0.674	−1	1	−1	
		−0.674–0.674	0	0	0	
		0.674–1.282	1	−1	1	
		1.282–1.645	2	−2	2	
		1.645–2.326	3	−3	3	
		2.326<	4	−4	4	
	Mortality	<−1.960		4	−4	4
		−1.960–−1.645		3	−3	3
		−1.645–−1.150		2	−2	2
		−1.150–−0.675		1	−1	1
		−0.675–0.675		0	0	0
		0.675–1.150		−1	1	−1
		1.150–1.645		−2	2	−2
		1.645–1.960		−3	3	−3
		1.960 <		−4	4	−4
HVI-NLM	Morbidity	<−1.960	−3	4	−4	
		−1.960–−1.645	−2	3	−3	
		−1.645–−1.150	−1	2	−2	
		−1.150–−0.675	0	1	−1	
		−0.675–0.675		0	0	
		0.675–1.150	0	−1	1	
		1.150–1.645	2	−2	2	
		1.645–1.960	3	−3	3	
		1.960 <	4	−4	4	
	Mortality	<−1.960			−4	4
		−1.960–−1.645			−3	3
		−1.645–−1.150			−2	2
		−1.150–−0.675			1	1
		−0.675–0.675			0	0
		0.675–1.150			1	
		1.150–1.645				
		1.645–1.960				
		1.960<				

## Data Availability

The data that support the findings of this study are available from Korea’s National Health Insurance Service; however, restrictions apply to the availability of these data, which were used under license for the current study and are not publicly available. However, the final analysis data are available from the authors upon reasonable request and with permission from the National Health Insurance Service. Please contact the author (Yeon-yeop Lim, siawase@kongju.ac.kr) if someone wants to request data from this study and submit the revised manuscript.

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
