# Peer review of "Elucidating Uncertainty in Heat Vulnerability Mapping: Perspectives on Impact Variables and Modeling Approaches"

_ijerph, 2024, doi:10.3390/ijerph21070815_

Round 1

Reviewer 1 Report

Comments and Suggestions for Authors

The authors of this manuscript present an interesting and useful study of HVI mapping techniques and comparison to health outcomes across South Korea. This study is well-framed, designed, and conducted, with clear explanations in the manuscript. In particular, I appreciate the good discussion of the implications of the study, particularly regarding the differences in policies between areas with high mortality and those with high morbidity. I have just a few recommendations that may improve the manuscript prior to publication.

First, regarding the variables selected for the HVI, would the number of doctors not be a positive indicator? I urge you to be careful with directionality. Increasing values in each of the other variables selected follow a convention of higher value = higher vulnerability. But higher doctors, I would argue, leads to less vulnerability. Should this be framed as the opposite, in which lower numbers of doctors = higher vulnerability?

Second, it has been my understanding that HVI mapping has traditionally been a technique used as proxy measures of potential health impacts when spatially comprehensive health data is not available. If you have health data or can model it in a spatially comprehensive way, as you did with mortality, what is the added benefit of the rest of the HVI, or in using HVI techniques at all? I’m not arguing that it isn’t a useful exercise, but I would like to see more of an explanation as to the specific benefits of HVI mapping above and beyond the health outcomes you’ve gathered or modeled.

Third, regarding limitations, I would recommend adding language describing the variable selection process. Simply selecting different (even similar) variables to include in the HVI may result in very different HVI maps (see Conlon et al. 2020), and therefore very different results. At least a brief mention of this would cover this significant caveat in the HVI creation process.

All in all, I believe this manuscript is almost ready for publication, so I recommend minor revision prior to publication. I provide more detailed notes on my recommended changes below.

References cited:

Conlon, K. C., Mallen, E., Gronlund, C. J., Berrocal, V. J., Larsen, L., & O’neill, M. S. (2020). Mapping human vulnerability to extreme heat: A critical assessment of heat vulnerability indices created using principal components analysis. Environmental Health Perspectives, 128(9), 1–14. https://doi.org/10.1289/EHP4030

 Gasparrini, A., Guo, Y., Hashizume, M., Lavigne, E., Zanobetti, A., Schwartz, J., … Armstrong, B. (2015). Mortality risk attributable to high and low ambient temperature: A multicountry observational study. The Lancet, 386(9991), 369–375. https://doi.org/10.1016/S0140-6736(14)62114-0

Detailed notes:

13 – recommend not using term “heat impact indicators” since indicators are often proxy measures in HVIs. Perhaps “health outcomes” instead?

22 – here as well, would recommend “outcomes” instead of “indicators” – this applies throughout the manuscript for clarity of terms with other HVI research.

147 – why have you selected daily maximum temperature? Other research has used daily average (e.g. Gasparrini et al. 2015). Maximum temperature may be valid as well, I would just like to see more explanation on this choice.

359 – I would recommend tempering this sentence, perhaps “confirmed” to “indicates” since this may not be the case in other contexts. For example, other HVI construction methods or other variable sets may produce differing results.

Author Response

We deeply appreciate the thoughtful comments provided by a reviewer and have implemented revisions accordingly. Any changes made to the manuscript are highlighted in red. Below, you'll find detailed responses addressing the reviewers' comments

Reviewer 2 Report

Comments and Suggestions for Authors

This is a review of the manuscript entitled "Elucidating Uncertainty in Heat Vulnerability Mapping: Perspectives on Impact Variables and Modeling Approaches", that has been submitted for its consideration for publication to IJERPH. The manuscript is aligned with the scope of the journal, but needs some modifications before its acceptance.

The introduction section is well-structured, but needs to include more references between lines 49-55 and 68-83. Expressions as "most other studies" or "previous studies" are used, but none are cited.

In the 2.1 section, please, use site literature to refer the Thiessen polygon method.

Figures in section 3.1 need to be clearer. The same for section 3.4. They are too difficult to read.

The discussion section is well conducted, but I believe that more comparison to other studies need to be made.

The conclusions section is too short and needs to be developed.

Comments on the Quality of English Language

First of all, the language needs a review, specially because there is too much passive voice in the text.

Author Response

We deeply appreciate the thoughtful comments provided by a reviewer and have implemented revisions accordingly. Any changes made to the manuscript are highlighted in red. Below, you'll find detailed responses addressing the reviewers' comments:

Round 2

Reviewer 2 Report

Comments and Suggestions for Authors

The comments that I addressed in the previous revision have been take into consideration. I would like to recommend the acceptance of this manuscript.